# Usability of Functional Electrical Stimulation in Upper Limb Rehabilitation in Post-Stroke Patients: A Narrative Review

**DOI:** 10.3390/s22041409

**Published:** 2022-02-12

**Authors:** Andreia S. P. Sousa, Juliana Moreira, Cláudia Silva, Inês Mesquita, Rui Macedo, Augusta Silva, Rubim Santos

**Affiliations:** 1Center for Rehabilitation Research—Human Movement System (Re)habilitation Area, Department of Physiotherapy, School of Health, Polytechnic of Porto, Rua Dr. António Bernardino de Almeida 400, 4200-072 Porto, Portugal; jmo@ess.ipp.pt (J.M.); claudiacostasilva78@gmail.com (C.S.); rmacedo@ess.ipp.pt (R.M.); afs@ess.ipp.pt (A.S.); 2Center for Rehabilitation Research—Human Movement System (Re)habilitation Area, Department of Functional Sciences, School of Health, Polytechnic of Porto, Rua Dr. António Bernardino de Almeida 400, 4200-072 Porto, Portugal; iam@ess.ipp.pt; 3Center for Rehabilitation Research—Human Movement System (Re)habilitation Area, Department of Physics, School of Health, Polytechnic of Porto, Rua Dr. António Bernardino de Almeida 400, 4200-072 Porto, Portugal; rss@ess.ipp.pt

**Keywords:** functional electrical stimulation, stroke, rehabilitation

## Abstract

Stroke leads to significant impairment in upper limb (UL) function. The goal of rehabilitation is the reestablishment of pre-stroke motor stroke skills by stimulating neuroplasticity. Among several rehabilitation approaches, functional electrical stimulation (FES) is highlighted in stroke rehabilitation guidelines as a supplementary therapy alongside the standard care modalities. The aim of this study is to present a comprehensive review regarding the usability of FES in post-stroke UL rehabilitation. Specifically, the factors related to UL rehabilitation that should be considered in FES usability, as well a critical review of the outcomes used to assess FES usability, are presented. This review reinforces the FES as a promising tool to induce neuroplastic modifications in post-stroke rehabilitation by enabling the possibility of delivering intensive periods of treatment with comparatively less demand on human resources. However, the lack of studies evaluating FES usability through motor control outcomes, specifically movement quality indicators, combined with user satisfaction limits the definition of FES optimal therapeutical window for different UL functional tasks. FES systems capable of integrating postural control muscles involving other anatomic regions, such as the trunk, during reaching tasks are required to improve UL function in post-stroke patients.

## 1. Introduction

Stroke is the major cause of disability [1,2]. It was estimated that after stroke, 70% of the patients present upper limb (UL) dysfunction, more than half present moderate to severe dysfunction, 40% are left with a non-functional arm with implications in quality of life [3,4] and only 5–20% recover UL function completely [5,6]. Although the main deficits were described for the contralesional limb (limb contralateral to the lesioned hemisphere), more recent studies have described postural control deficits also in the ipsilesional limb (limb ipsilateral to the lesioned hemisphere) [7,8,9], which were demonstrated to interfere in the rehabilitation of contralesional UL function [7]. Considering the determinant role of UL during activities of daily living (ADL) [10,11,12,13], the rehabilitation of UL function, namely to improve the ability to reach and grasp, required in over 50% of ADLs [14], is a primary aim in stroke rehabilitation.

It is expected that stroke will continue to be the main cause of disability-adjusted life years until 2030. Consequently, rehabilitation care will continue to represent an important and growing burden for the global health system difficulting the ideal of personalized rehabilitation based on a ratio of one therapist for one patient [15,16,17]. The functional electrical stimulation (FES), combining electrical stimulation with the performance of functional tasks [18], could be a strategy to extend the UL rehabilitation beyond health care units such as the home setting or other environments selected by the user [19,20]. By being used beyond health care units, FES has the potential of alleviating the pressure on existing health care infrastructures but may also constitute a powerful motivational factor since it is performed in a more functional context [21].

Previous systematic reviews with meta-analysis have demonstrated the positive effects of FES on ADL outcomes [19,20,22]. This modality was even recommended by the recent Clinical Guidelines for Stroke Management [23] and Guidelines for Adult Stroke Rehabilitation and Recovery [17] as a supplementary therapy alongside standard care modalities. However, according to the data gathered by previous systematic reviews [19,20,22], the studies included did not assess the quality of movement as an outcome to assess the role of FES in UL rehabilitation of post-stroke patients. In this context of the applicability of the FES as a tool that contributes to enhancing the quality of movement of the post-stroke subject, it is important to consider the concept of usability as a determining factor. Usability is defined as a measure that evaluates the user’s performance with a specific product or system, in particular with regards to the quality of the user’s experience, and can be analyzed through effectiveness and user satisfaction [24].

The aim of this study is to present a comprehensive review regarding the usability of FES in UL rehabilitation of post-stroke subjects. Specifically, the factors related to post-stroke UL rehabilitation that should be considered in FES usability, as well a critical review of the outcomes used to assess FES usability, are presented. Accordingly, this review is divided into four sections. In the present section, the epidemiological data that sustain the need of exploring strategies to improve UL function in post-stroke patients is presented. The second section presents an overview of the particularities of UL rehabilitation in post-stroke subjects and its implications in FES usability. Section three is focused on FES usability and gathering and discussing the available measures to assess UL movement quality indicators. This section presents a critical review of the outcomes to be considered to assess FES effectiveness. Finally, the last section reports the final considerations of this review. Figure 1 summarize the main arguments that highlight the need to review FES usability in UL rehabilitation in post-stroke patients.

## 2. UL Post Stroke Rehabilitation Factors That Should Be Considered in FES Usability

The goal of rehabilitation is the reestablishment of pre-stroke motor stroke skills by stimulating neuroplasticity [25], a complex combination of spontaneous and learning-dependent motor processes including restitution, substitution and compensation [26]. Evidence suggests that post-stroke recovery is largely dependent on learning adaptation strategies, particularly those that promote the organization of residual neural circuits and/or allow access via new and alternative pathways [27].

Neuromuscular electrical stimulation (NMES) has emerged as an efficient tool to induce activity-dependent plasticity in neural circuits in stroke patients [28,29,30]. NMES consists of a series of intermittent electrical stimuli applied over the muscle or the nerve trunk to elicit tetanic muscle contractions [31,32]. When used in an isolated way, this modality is mainly used for: (1) maintaining/preserving neuromuscular function during disuse; (2) restoring neuromuscular function after disuse; and (3) improving neuromuscular function in healthy individuals, including athletes [31]. In stroke patients, NMES is used to assist voluntary movements during functional tasks, being commonly designated as FES [33]. In fact, FES consists of the application of moderate-intensity and cyclic electrical stimulation over selected muscles to generate functional movements that mimic voluntary contractions and to restore functions that were lost [19,32]. The two common uses of FES are to replace function (i.e., as an orthotic device) and to retrain function (i.e., as a therapeutic device) [34]. For more detailed information about FES systems used in post-stroke rehabilitation, the review performed by Marquez-Chin and Popovic [32], 2020, should be consulted. The main differences in the advantages of NME and FES are summarized in Figure 2.

Based on the arguments presented in Figure 2, technology-assisted training of arm-hand skills based on FES is considered an attractive rehabilitation option. Particularly because it has the ability to provide intensive periods of experience of the right sequence and magnitude of muscle activity to perform functional tasks to facilitate motor re-learning [35] with relatively low demand on human resources [3]. In fact, it is well known that among the many endogenous and exogenous events that may trigger the post-injury neural plasticity phenomena, experience with repetition is a strong modulator of cortical structure and function [26,36,37,38]. Moreover, task-specific training in the familiar contexts of the patients has the potential to enhance the acquisition of similar behaviors by transference [26,38,39].

A wide body of evidence supports the ability of FES in improving motor control [40,41,42]. This was demonstrated in output related variables such as range of movement, strength and postural tone [40,41,43], but also in central processing related variables [44,45,46,47,48,49,50]. Specifically, it was shown that FES leads to activation of the contralateral primary motor control somatosensory cortex and bilateral supplementary motor areas and prefrontal cortex [44,45,46,47] and a modification of corticospinal excitability and the related output [48,49,50]. The neuroplastic changes are greater if the practice method is meaningful, repetitive and intensive in nature [43,51,52]

Based on the exposure and considering that the motor learning is specific and depends on the repetition of the motor task, its novelty and concurrent volitional effort, the FES must provide the execution of the movement for functional tasks performance with precision and efficiency [53]. In fact, assistive technologies, including FES, together with task-orientated training, combines two rehabilitation paradigms for the UL, providing a means to enable patients to practice meaningful, functional tasks more intensely and more effectively on their own by increasing neuronal functional connectivity as represented in Figure 3.

As mentioned in Figure 3, several cortical areas are frequently lesioned in stroke patients. The middle cerebral artery territory is most affected [54]. In these patients, there is a high probability for the dysfunction of the cortico-reticular pathway that enables the connection between the cerebral cortex, mostly in area 6, including the premotor cortex, and the supplementary area, to regulate the coordination between postural and movement control [55,56]. Therefore, to interfere in functional disability, FES should be used not only in movement-related muscles such as deltoids, triceps and the wrist and finger extensors/flexors [19] but also in postural control-related muscles in different UL tasks [57].

During reaching tasks, each UL movement is preceded (preparatory anticipatory postural adjustments) and accompanied (accompanying anticipatory postural adjustments) by anticipatory adjustments that must also occur in the contralateral side to movement execution [58,59,60,61], i.e., when movement occurs in the contralesional side, the ipsilesional side has a key role in ensuring the proper postural background for movement efficiency. Therefore, the incorporation and adaptation of FES technology for enhancing postural activity during reaching would increase its potential for improving UL movement quality and function. The evidence demonstrates that post-stroke subjects with lesions in the middle cerebral territory also present impairments in the ipsilesional side, mainly related to postural control, which are even more important to this adaptation [7]. Since shoulder and elbow training only improves motor impairment in the shoulder and elbow [62], and that training of the wrist and finger extensors only improves hand function [63], FES systems should have the capacity of stimulating postural control muscles involving other anatomic regions, linking the trunk and ipsilesional side to improve UL function. In fact, evidence shows that the benefits of FES are greatest when UL muscles are trained in a synergic pattern [62,63,64,65]. Hence, it is important for an FES system to: (1) accurately assist functional tasks in a synergic way; (2) encourage user effort; and (3) ensure muscle recruitment selectivity [66]. To achieve these assumptions, the FES systems should include the selection of a set of parameters capable of providing functional movements.

It is known that square or rectangular biphasic pulse shape is more efficient for nerve stimulation due to an instantaneous increase in current to the maximal level [67]. The pulse frequency, typically ranging between 15 and 40 Hz, affects the type of muscle contraction and the level of force produced [32,67,68]. The higher stimulus frequencies generate higher forces (temporal summation) but produce fatigue of the muscle fiber and a rapid decrease in contractile force [32,33,69]. An optimal system uses the minimum stimulus frequency, producing a fused response that, in cases of upper limb applications, ranges from 12–16 Hz [70]. The strength of a muscle contraction may also be increased by increasing the number of motor units activated (spatial summation). This is achieved by increasing the stimulus pulse amplitude and/or pulse duration, which effectively increases the electric charge injected, producing a larger electric field and broader region of activation so that more axons and motor units are activated [5,69,71]. The pulse interval typically ranges between 200–400 microseconds [5,69]. It was demonstrated that high intensity and large pulse durations increase the excitability of corticomotor projections to stimulated muscles [66]. These parameters combined with a personalized adjustment of the duty cycle can increase the self-perception of improvement of UL function in post-stroke patients during turning on the light and drinking tasks [72,73]. However, outcome measures considered in most previous studies, despite demonstrating a positive effect over ADL [19,20,22], include qualitative measures that induce a certain level of subjectivity. Consequently, more studies are required to establish conclusions about the FES optimal therapeutic window for UL rehabilitation in post-stroke patients, particularly in other UL tasks that are not yet assessed [19].

Apart from the limitations previously mentioned, technological advances were made to improve muscle recruitment selectivity. From these, the development of the multifield electrode system enhances the generation of a localized electric field that increases the selectivity in motor units recruitment with reduced discomfort and fatigue [74], compared to other methods of electrical stimulation [18,75]. Multifield FES systems are developed for various applications [74,76]. Some studies have already proven their effectiveness regarding grasp recovery in post-stroke patients [74,76]. This was ensured by optimizing the shape, position and size of the stimulation surface on the forearm and minimizing the difference between the desired movement and that generated by FES [74,76].

The performance of functional movements resulting from the selective and synergistic activation of muscle groups associated with multifield FES depends on the adequate positioning and calibration of the electrodes, namely in terms of amplitude, pulse width and frequency [75,77]. Multifield FES provides this selective synergistic muscle activation while respecting neuroanatomical variability [75] to optimize the functional movement of the contralesional UL [77]. This can be achieved as a consequence of:the possibility of desynchronized activation of different portions of the muscle [74];the possibility of individual adjustment of the stimulation location [77];the possibility of varying the stimulation parameters and patterns of stimulation, namely frequency, amplitude, pulse duration and stimulation channel, to recruit the more adequated synergy [77].

Previous studies concerning FES multifield systems have demonstrated the stimulation zones and related parameters that lead to selective activation of the common extensor of the fingers, extensor digiti minimi, cubital extensor, radial extensor of the carpus, abductor pollicis longus, extensor pollicis brevis and extensor pollicis longus [75,77]. The results obtained by more recent studies demonstrate that a protocol based on an individual adjustment of the mentioned pre-defined transcutaneous stimulation zones allows consistency between intervention sessions in post-stroke patients [72,73]. The advantages of multifield FES against more traditional approaches are summarized in Figure 3.

## 3. FES Usability in Post-Stroke Patients

Concerning technological devices developed for people with limitations in activity and participation, such as post-stroke patients, the usability tests become an essential tool to ensure that a product has the desired impact. To evaluate FES usability, the quality of the user’s experience in terms of satisfaction and effectiveness should be considered [23]. Usability tests are designed to evaluate the product under controlled conditions, simulating the interaction and the quality of the user experience [78,79].

### 3.1. FES User Satisfaction

Satisfaction is the users’ comfort with and positive attitudes towards the use of a system [80]. The System Usability Scale [81] and Quebec User Evaluation of Satisfaction with Assistive Technology questionnaire [81,82] were used to assess the usability and the related satisfaction of both the therapists and subjects for lower limb multifield technology. The results of these studies demonstrate that it is feasible to include surface multifield technology while keeping a device simple and intuitive for successful integration in common neurorehabilitation programs. The Patient Global Impression of Change was used to assess the post-stroke patients’ perception of change concerning UL movement when this was assisted by multifield FES [72,73]. The average improvement described by the participants ranged between “somewhat better” and “moderately better”.

### 3.2. FES Effectiveness

Effectiveness is the accuracy and completeness with which users achieve certain goals. Indicators of effectiveness include quality of solution and error rates [80].

#### 3.2.1. Clinical Measures

To analyze FES effectiveness, clinical measures related to ADL, functional recovery measures and muscle-related outcomes were considered (Table 1) [19,20,22]. It was demonstrated that FES is effective in improving ADL, expressed through Functional Independence Measure and Upper Extremity Function Test scores, and in functional motor recovery, through a Fugl-Meyer Assessment score [19]. However, its effectiveness was not demonstrated in the other clinical measures, which need to be discussed.

When analyzing the results provided by the tools presented in Table 1, several aspects should be considered. In addition to observer bias, it should be considered in the hypothesis that these instruments could not be sensitive enough to detect improvement signs regarding complex UL motor function [83]. Furthermore, the tools cannot explain the underlying biomechanical characteristic of motor function deficits, and their scores alone do not clarify whether the observed changes depend on true recovery or compensatory strategies [83]. In fact, several compensatory strategies expressed through pathological synergies were described for post-stroke patients during reaching tasks as a consequence of the available motor strategies [84]. In other words, to compensate for upper limb impairment, patients tend to recruit alternative strategies to improve functional arm and hand use. The neurophysiologic explanation for this phenomenon highlights the post-trauma nervous system’s ability to exploit the motor system’s redundancy by replacing lost motor pattern elements with new ones to achieve the desired task [85]. In fact, it is well known that after a lesion, the nervous system can be reorganized, producing an adaptive or maladaptive sensoriomotor behavior, thus highlighting the importance of cortical reorganization through selective afferent input to optimize internal representation and influence movement control (Figure 3) [26,86]. Moreover, considering that most stroke lesions occur in the territory of the middle cerebral artery, presenting a high probability of damage of pathways with predominant ipsilesional disposition mainly related to postural control, the tools used to assess UL function should also measure bilateral postural control dysfunction [7]. Unfortunately, both research and clinical rehabilitation involving post-stroke subjects is focused on contralesional side impairments, while ipsilesional impairments are attributed to an adaptative strategy [7]. Based on this, the instruments regarding UL function are used to reference the ipsilesional side (i.e., Stroke Rehabilitation Assessment of Movement Measure [87], Motor Assessment Scale [88], Chedoke–McMaster Stroke assessment [89]). This approach has negative consequences in the decision-making process since: (1) it limits the identification of a possible ipsilesional impairment related to postural control; (2) it limits improvement since movement failure of the contralesional side is also related to ipsilesional postural control dysfunction; (3) it compromises the inter-limb coordination necessary for most functional activities [7].

#### 3.2.2. Laboratory Measures

Laboratory measures allow an accurate and objective assessment of FES effectiveness regarding the selectivity and quality of UL movement. Some authors [90,91,92,93,94,95] used electromyography to analyze reaction times and activation magnitudes of sustained muscle contractions, as well as to calculate ratios between agonist and antagonist muscle activation. Gripping power was also assessed by dynamometry in another study [96]. However, although these measures allow a more objective knowledge of motor function, they still do not analyze the quality of movement. 

Quality of movement can be accurately evaluated by kinematic analysis. In fact, in the last decade, the kinematics of the ULs of neurological patients, mostly after stroke, were studied in order to quantify movement objectively [97]. According to Ozturk et al. [98], this analysis depends on four major factors: (a) motion capture systems, (b) movement category, (c) kinematic metrics extracted and (d) interpretation of these kinematic metrics.

For this purpose, the most widely used type of motion capture system is the optoelectronic system with passive markers [83], which is the golden standard in kinematic analysis because of its high accuracy and reliability [97,98,99]. This system uses retro-reflective markers (passive or active) in which absolute position is detected by multiple video cameras in relation to a reference position [99]. Although portable and markerless systems, such as inertial or electromagnetic systems, appear to be promising alternatives for the kinematic analysis of the ULs in stroke patients, the literature proving its validity for this purpose is scarce [99,100,101].

The motor tasks generally used to study the function of ULs can be categorized into functional movements (reaching movements and path drawing) and ADL [97,98,102], as proposed by van Tuijl et al., 2002 [102]. Although several authors [97,103,104,105] defend the analysis of goal-oriented tasks, such as performing an ADL, to increase the validity of studies, half of the studies still analyze functional movements [106]. Within the ADL category, the most performed task is drinking [106]. This seems to be a rich task for the kinematic analysis of the UL as it includes sub-tasks such as reaching, grasping, transporting and manipulating an object, which makes possible the study of these different motor skills [106]. However, it may become too complex for subjects with moderate or severe impairment, which could decrease the number of participants in these studies [106]. Therefore, simpler ADLs are recommended to include subjects with more severe impairment and increase the number of participants [106]. Figure 4 represent two ADL tasks with different levels of motor control complexity.

Numerous kinematic metrics were used in the evaluation of UL movement in post-stroke patients [83,107], which may be related to the lack of clarity regarding the ULs motor planning [108]. Based on the theories of UL movement planning [108], kinematic metrics can be classified into two categories: end-point (hand or wrist) kinematic metrics and joint kinematic metrics [97,108]. End-point kinematic metrics are widely calculated by 3D Cartesian coordinates of only one marker on the wrist (or hand) and analyze different characteristics of movement, such as speed, efficiency, smoothness and control strategy [97]. The most analyzed are “time to complete the task” [103,104,109,110,111,112,113], “peak velocity” [98,103,107,110,111,113] and the “number of peaks in velocity profile” [103,104,107,112,113]. Joint kinematic metrics include joint angles [103,105,107,111], angular velocities [103,104,105], inter-joint coordination between shoulder and elbow [98,103,107] and trunk displacement [98,103,104,110,112] (which is also used to quantify compensatory strategies).

The interpretation of kinematic variables is unclear. Subramanian et al. [114] suggested the association between end-point kinematics and motor performance, as well as between joint kinematics and movement quality. Subramanian et al. [114], and other authors [98], also suggested that movement quality kinematics are more sensitive in identifying UL deficits, while others [103,113] have argued that motor performance kinematics are sensitive to change over time and discriminate healthy subjects from those with stroke, as well as subjects with moderate impairment from those with mild impairment. Murphy et al. [112] also speculate that some metrics, such as trunk displacement, primarily reflect the component of compensation, and others, such as movement smoothness, the recovery. However, these associations and their meanings are not well established [83]. Similarly, and also for these variables, the role of FES application in the ipsilesional side should be considered, as according to our knowledge, no study has considered the impairments already demonstrated in the ipsilesional side in stroke patients in the application of FES [8,9,115]. The metrics that should be considered to assess the influence of FES in movement quality indicators are summarized in Figure 5.

## 4. Concluding Remarks

FES seems to be a promising tool for improving UL function in post-stroke patients. However, the lack of studies evaluating motor control variables, specifically movement quality indicators, combined with user’s satisfaction, limits the establishment of conclusions regarding its usability. This lack of information compromises the establishment of the optimal therapeutical window for different UL functional tasks for post-stroke patients. Besides considering the stimulation parameters that best assist the desired movement, the optimal therapeutical window should also consider the muscle synergy that needs to be recruited. FES systems capable of integrating postural control muscles involving other anatomic regions, such as the trunk and ipsilesional side, are required to improve UL function in post-stroke patients.

## Figures and Tables

**Figure 1 sensors-22-01409-f001:**
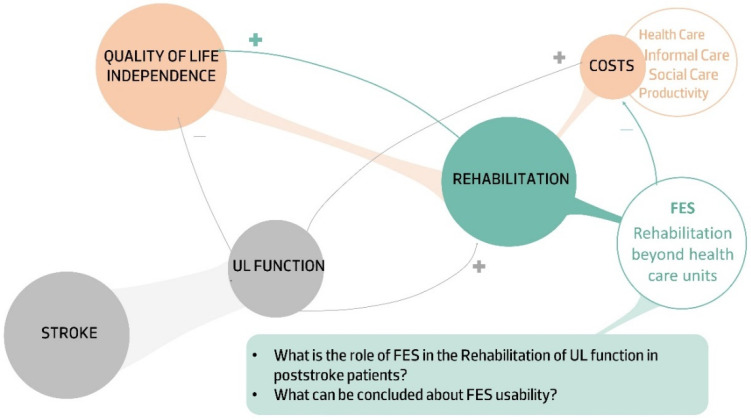
Summary of the main arguments that sustain the use of FES as a strategy to improve UL function.

**Figure 2 sensors-22-01409-f002:**
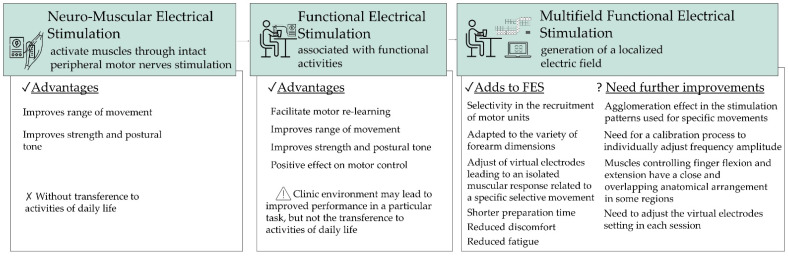
Advantages of NMES and FES-based systems.

**Figure 3 sensors-22-01409-f003:**
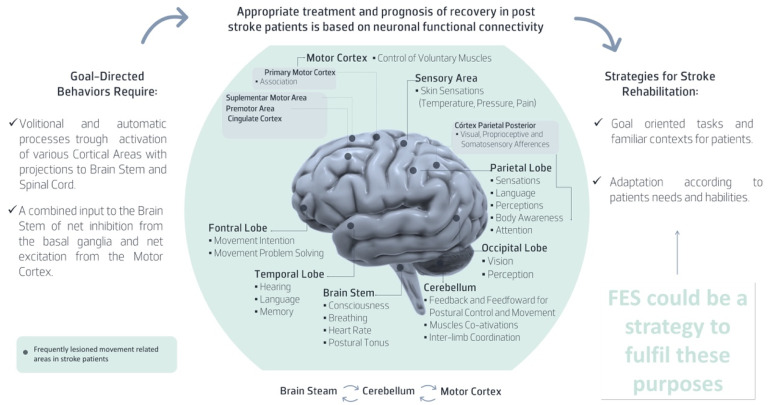
Demonstration on how FES could increase neuronal functional connectivity in post-stroke patients.

**Figure 4 sensors-22-01409-f004:**
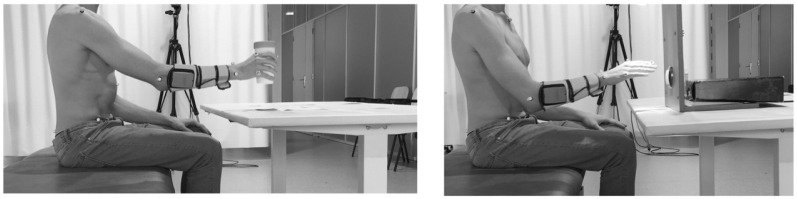
Representation of two tasks of different levels of motor control complexity assisted by multifield FES, drinking task (on the **left**) and turn on the light tasks (on the **right**).

**Figure 5 sensors-22-01409-f005:**
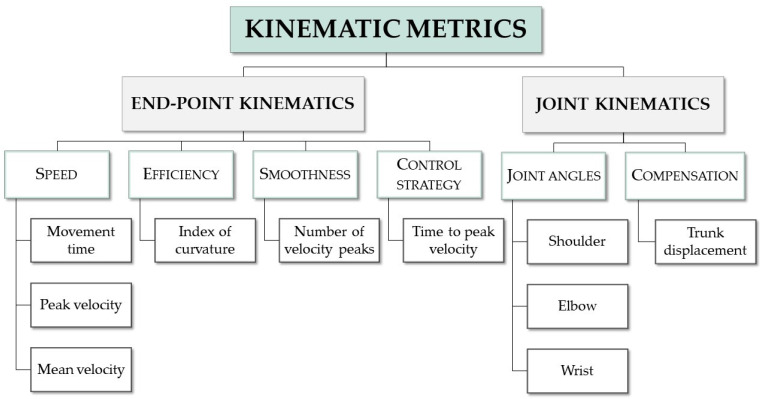
Summary of kinematics measures that should be considered to assess the influence of multifield FES in UL movement quality.

**Table 1 sensors-22-01409-t001:** Clinical measures used to assess FES effectiveness.

Clinical Measure	Tool
ADL	Functional Independence Measure Upper Extremity Function Test Arm Motor Ability Test Chedoke Arm and Hand Activity Inventory Functional Independence Measure Upper Extremity Function Test
Funcional Motor Recovery	Motor Assessment Scale Hand Movements Motor Assessment Scale Upper Arm Function Fugl-Meyer Assessment Box and Block Test Action Research Arm Test Functional Test for the Hemiparetic Upper Extremity Functional Test for the Hemiparetic Upper Extremity Chedoke McMasters Stroke Assessment Nine Hole Peg Test Ten Cup Moving Test
Muscle related	Modified Ashworth Scale Force

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
