# Peer review of "Usability of Functional Electrical Stimulation in Upper Limb Rehabilitation in Post-Stroke Patients: A Narrative Review"

_sensors, 2022, doi:10.3390/s22041409_

Round 1

Reviewer 1 Report

The objective is impactive and constructive if done correctly. FES is getting more and more attention in upper extremity rehabilitation with the emergence of the Brain-computer interface-driven FEC and neurofeedback training protocols.

Major concerns:

1- the manuscript is not wel strucutred. There should be a scientific methodology on how the literature review is conducted. Although the objective is to investigate the effect of FES on UE through a literature review of the previous publications, however, it is hard to follow the objective within the manuscript. I would introduce the structure of the manuscript in the last paragraph of the introduction. 

2-You should also introduce how you picked the citation you chose for this review. 

3- The pictures are low quality.

4- The introduction does not establish the significance of FES on upper extremity rehabilitation. 

5- Many of the term used in the manuscript including the FES is not well defined in the appropriate section. 

6- There are not enough quantitative values in the manuscript for the audience to assess the effectiveness and efficacy of FES.

Author Response

  1. i) Response to comments by Reviewer #1

We are grateful to the Reviewer for his/her comments and suggestions that have helped us improving our work. We have taken into account the comment made. Below we itemize and provide more detailed explanations of these changes.

Please note that the changes that we have made are highlighted in the revised version.

General comments

The objective is impactive and constructive if done correctly. FES is getting more and more attention in upper extremity rehabilitation with the emergence of the Brain-computer interface-driven FEC and neurofeedback training protocols.

Authors: We would like to thank the reviewer for the positive comment.

Major concerns:

1- the manuscript is not wel strucutred. There should be a scientific methodology on how the literature review is conducted. Although the objective is to investigate the effect of FES on UE through a literature review of the previous publications, however, it is hard to follow the objective within the manuscript. I would introduce the structure of the manuscript in the last paragraph of the introduction.

Authors: We have introduced a paragraph in the introduction explaining the structure of the review and manuscript. To clarify the structure of the manuscript we have included a new figure. We have also revised the whole manuscript to clarify the organization of the manuscript.

2-You should also introduce how you picked the citation you chose for this review.

Authors: We have clarified in the manuscript that this manuscript is not a systematic review, but a critical review about the role of FES in upper limb rehabilitation of post-stroke subjects integrating concepts of effectivity, biomechanics and neuroscience.

3- The pictures are low quality.

Authors: We have improved the quality of the figures. Also, because we are afraid, they lose quality as they were initially submitted in the PDF format only, in the resubmission of the revised manuscript we also include the figures in a separated file.

4- The introduction does not establish the significance of FES on upper extremity rehabilitation.

Authors: We have revised the introduction session to clarify this aspect.

5- Many of the term used in the manuscript including the FES is not well defined in the appropriate section.

Authors: The definition of FES was improved and transferred to introduction. Additionally, other concepts were defined as usability, satisfaction and effectiveness, contralesional and ipsilesional side of the stroke and multifield FES were differed along the manuscript.

6- There are not enough quantitative values in the manuscript for the audience to assess the effectiveness and efficacy of FES.

Authors: We have included more detailed information regarding this aspect as requested.

Reviewer 2 Report

Peer-review of the manuscript: ”The role of functional electrical stimulation in the rehabilitation of upper limb function in poststroke patient”

  1. General remarks:

The authors achieved a literature review that aimed to ”present a critical (? – but not systematic, following a related standardized methodology, for instance: PRISMA ?; I see that this manuscript is categorized by the Editors as ”Comprehensive Review”; so, if according to the editorial policy of this journal there are accepted literature reviews elaborated without following a standardized methodology for searching, filtering and selecting the afferent bibliographic resources, i. e. fulfilled in a rather narrative way, then this manuscript is OK from this point of view – o. n.) review about the role of Functional Electrical Stimulation (FES) in UL function of post-stroke patients and the main reasons that limit the establishment of conclusions about its effectiveness.”

- in what consists of the important added value brought by this manuscript to the domain approached, compared, for instance, to a quite recent article with similar focus – but which is a related systematic review and meta-analysis, cited by the authors (” Eraifej J, Clark W, France B, Desando S, Moore D. Effectiveness of upper limb functional electrical stimulation after stroke for the improvement of activities of daily living and motor function: a systematic review and meta-analysis. Syst Rev. 2017 Feb 28;6(1):40. doi: 10.1186/s13643-017-0435-5. PMID: 28245858; PMCID: PMC5331643.”) ? – if possible, the authors should highlight this aspect

  1. Specific remarks and suggestions:

- the authors should present the differences – conceptually and practically – between the two types of Neuro-Muscular Electrical Stimulation (NMES): for motor re-learning and for – including partly assistive – FES (Sheffer LR, Knutson JS, Chae J – Therapeutic Electrical Stimulation, in: Frontera WR, DeLisa JA, Gans BM et al. (Eds.): DeLisa’s Physical Medicine & Rehabilitation Principles and Practice, Fifth Edition, Vol. II, Wolters Kluwer Health. Lippincott Williams & Wilkins, Philadelphia, USA, 2010;       Knutson JS, Sheffer LR, and Chae J – Functional neuro-muscular Electrical Stimulation, In: Frontera, W.R. and DeLisa, J.A. (eds.) DeLisa’s Physical Medicine & Rehabilitation Principles and Practice, Woltres Kluwer Health. Lippincott Williams & Wilkins, Philadelphia, USA, 2010)

- the authors should specify whether the applied electrical stimuli consisted of sinusoidal (i. e. with alternating positive and negative waves, resulting thus in cycles) or only of positive half waves/ (redressed) impulses ? – because, rigorously, only the alternating/ sinusoidal waves (having thus cycles) are measured in Hertz (Hz) – see, in this respect, the bibliographic reference no. 43. On the other hand, regarding the electrical stimulation parameters, the authors mention the related, quoted by them, references (42, 44 and 45) – these are not available in extenso, in order to be verified

- Figg. 1 and 3 are of low resolution, the text inside them is very hardly legible/ readable

- Fig. 2: there is not specified the origin of the two images within it (are them from the own – of the authors – casuistry or they are reproduced from the literature; in this later situation it has to be specified their original provenance ad the acceptance for reproduction from those having the intellectual property of the respective images).

In conclusion: if, as asserted above, the editorial policy of this journal accepts literature reviews elaborated without following a standardized methodology for searching, filtering and selecting the afferent bibliographic resources (i. e. fulfilled in a rather narrative way), this manuscript is OK from this point of view, being thus acceptable with minor revisions. But, if this is not the case, then this manuscript needs major revision.   

Author Response

  1. ii) Response to comments by Reviewer #2

We are grateful to the Reviewer for his/her comments and suggestions that have helped us improving our work. We have taken into account the comment made. Below we itemize and provide more detailed explanations of these changes.

Please note that the changes that we have made are highlighted in the revised version.

General remarks

The authors achieved a literature review that aimed to ”present a critical (? – but not systematic, following a related standardized methodology, for instance: PRISMA ?; I see that this manuscript is categorized by the Editors as ”Comprehensive Review”; so, if according to the editorial policy of this journal there are accepted literature reviews elaborated without following a standardized methodology for searching, filtering and selecting the afferent bibliographic resources, i. e. fulfilled in a rather narrative way, then this manuscript is OK from this point of view – o. n.) review about the role of Functional Electrical Stimulation (FES) in UL function of post-stroke patients and the main reasons that limit the establishment of conclusions about its effectiveness.”

- in what consists of the important added value brought by this manuscript to the domain approached, compared, for instance, to a quite recent article with similar focus – but which is a related systematic review and meta-analysis, cited by the authors (” Eraifej J, Clark W, France B, Desando S, Moore D. Effectiveness of upper limb functional electrical stimulation after stroke for the improvement of activities of daily living and motor function: a systematic review and meta-analysis. Syst Rev. 2017 Feb 28;6(1):40. doi: 10.1186/s13643-017-0435-5. PMID: 28245858; PMCID: PMC5331643.”) ? – if possible, the authors should highlight this aspect

Authors: We confirm that this manuscript is not a systematic review but a narrative review. In the instructions for authors this kind of manuscript is enabled (Reviews: These provide concise and precise updates on the latest progress made in a given area of research). We have also clarify in the introduction section what this review adds in relation to the previous systematic reviews published in the area.

Specific remarks and suggestions:

- the authors should present the differences – conceptually and practically – between the two types of Neuro-Muscular Electrical Stimulation (NMES): for motor re-learning and for – including partly assistive – FES (Sheffer LR, Knutson JS, Chae J – Therapeutic Electrical Stimulation, in: Frontera WR, DeLisa JA, Gans BM et al. (Eds.): DeLisa’s Physical Medicine & Rehabilitation Principles and Practice, Fifth Edition, Vol. II, Wolters Kluwer Health. Lippincott Williams & Wilkins, Philadelphia, USA, 2010;       Knutson JS, Sheffer LR, and Chae J – Functional neuro-muscular Electrical Stimulation, In: Frontera, W.R. and DeLisa, J.A. (eds.) DeLisa’s Physical Medicine & Rehabilitation Principles and Practice, Woltres Kluwer Health. Lippincott Williams & Wilkins, Philadelphia, USA, 2010)

Authors: We have clarified these issues in figure 3.

- the authors should specify whether the applied electrical stimuli consisted of sinusoidal (i. e. with alternating positive and negative waves, resulting thus in cycles) or only of positive half waves/ (redressed) impulses ? – because, rigorously, only the alternating/ sinusoidal waves (having thus cycles) are measured in Hertz (Hz) – see, in this respect, the bibliographic reference no. 43. On the other hand, regarding the electrical stimulation parameters, the authors mention the related, quoted by them, references (42, 44 and 45) – these are not available in extenso, in order to be verified

Authors: We have clarified that the applied stimulus involved biphasic symmetric waves. We have also corrected the citations.

- Figg. 1 and 3 are of low resolution, the text inside them is very hardly legible/ readable

Authors: We have improved the quality of the figures. Also, because we are afraid they lose quality as they were initially submitted in the PDF format only, in the resubmission of the revised manuscript we also include the figures in a separated file.

- Fig. 2: there is not specified the origin of the two images within it (are them from the own – of the authors – casuistry or they are reproduced from the literature; in this later situation it has to be specified their original provenance ad the acceptance for reproduction from those having the intellectual property of the respective images).

Authors: We confirm that all the images are of the property of the authors.

In conclusion if, as asserted above, the editorial policy of this journal accepts literature reviews elaborated without following a standardized methodology for searching, filtering and selecting the afferent bibliographic resources (i. e. fulfilled in a rather narrative way), this manuscript is OK from this point of view, being thus acceptable with minor revisions. But, if this is not the case, then this manuscript needs major revision.

Authors: All the points mentioned by the Reviewer were addressed excepting the design of the study that was maintained as a narrative review respecting the instructions for authors.

Round 2

Reviewer 1 Report

1-The quality of the figures is still low.

Line 94: as not "has". Please check the document for similar errors.

2-line 105 what resources are you talking about. 

3-line 138 and 148: you are providing a lot of important information here without any reference to back them up, such as: "the width of the biphasic symmetric pulse determines the number of muscle fibers that are activated"; "that the pulse frequency determine the speed at which the stimulus impulses are applied to the muscle"; "The minimum stimulus frequency that generates a fused muscle response is proximally 12.5 Hz." each of these statements need a reference. 

4- Avoid long sentences. E.g., L 168 to 173 should be divided into a couple of sentences. 

5- Please specifically mentioned the objective of your review and the summary of your conclusion in the abstract. 

6-Figure3: Please clarify the difference between NMES and FES in the text and provide examples of the applications of each for UL rehabilitation in the stroke population.

Overall as you mentioned in the introduction the aim of your study is "to present the critical review about the role of FES in the rehabilitation of UL function in post-stroke patients"; I do not know how this is possible without providing a single quantitative value especially in section 2. If the objective is to provide the guideline for conducting an FES study there many other factors like the study designs, the cointerventions, training protocol,.....in addition to the outcome measure, so, it is currently confusing, what the manuscript is trying to accomplish. Most of the focus on section two is stimulation parameters and selective stimulation and then you move on to the critical review of the outcome measures. without reviewing any other aspect of the FES studies. 

There is valuable information in this submission especially in section 2 but the manuscript does not clearly follow the objective. 

Author Response

  1. i) Response to comments by Reviewer #1

We are grateful to the Reviewer for his/her comments and suggestions that have helped us improving our work. We have taken into account the comment made. Below we itemize and provide more detailed explanations of these changes.

Please note that the changes that we have made are highlighted in the revised version.

Comments

1-The quality of the figures is still low.

Authors: We have improved the quality of the figures.

Line 94: as not "has". Please check the document for similar errors.

Authors: We have corrected this typo and other similar in the whole manuscript.

2-line 105 what resources are you talking about.

Authors: We have clarified that we were referring to human resources.

3-line 138 and 148: you are providing a lot of important information here without any reference to back them up, such as: "the width of the biphasic symmetric pulse determines the number of muscle fibers that are activated"; "that the pulse frequency determine the speed at which the stimulus impulses are applied to the muscle"; "The minimum stimulus frequency that generates a fused muscle response is proximally 12.5 Hz." each of these statements need a reference.

Authors: we have included the references supporting the presented data.

4- Avoid long sentences. E.g., L 168 to 173 should be divided into a couple of sentences.

Authors: We have revised the whole manuscript and revised long sentences as the one mentioned by the reviewer.

5- Please specifically mentioned the objective of your review and the summary of your conclusion in the abstract.

Authors: We have revised the abstract section to include the objective and the summary of the conclusion.

6-Figure3: Please clarify the difference between NMES and FES in the text and provide examples of the applications of each for UL rehabilitation in the stroke population.

Authors: We have included the information requested.

Overall as you mentioned in the introduction the aim of your study is "to present the critical review about the role of FES in the rehabilitation of UL function in post-stroke patients"; I do not know how this is possible without providing a single quantitative value especially in section 2. If the objective is to provide the guideline for conducting an FES study there many other factors like the study designs, the cointerventions, training protocol,.....in addition to the outcome measure, so, it is currently confusing, what the manuscript is trying to accomplish. Most of the focus on section two is stimulation parameters and selective stimulation and then you move on to the critical review of the outcome measures. without reviewing any other aspect of the FES studies.

Authors: We would like to thank the Reviewer for the comment as it helped us understanding that objective of the manuscript was wrongly formulated. We agree with the Reviewer that the information provided is not enough to stablish conclusions about the role of FES in the rehabilitation of UL function. This wasn´t the real objective since this was already assessed in previous systematic reviews. The objective of session 2 was to provide an overview of the particularities of UL rehabilitation in poststroke subject that should be considered in FES usability assessment. We have revised the whole manuscript to clarify this aspect.

There is valuable information in this submission especially in section 2 but the manuscript does not clearly follow the objective.

Authors: We would like to thank the Reviewer the positive comment. As we mentioned in the previous comment, we have clarified the objective of the manuscript and have revised the whole manuscript accordingly to clarify the alignment between the review and the purpose.